# Modeling Wicking Fabric Inhibition Effect on Frost Heave

**Yuwei Wu [1], Tatsuya Ishikawa [2,\*], Kimio Maruyama [3], Chigusa Ueno [3], Tomohisa Yasuoka [1] and Sho Okuda [1]**

[1] Graduate School of Engineering, Hokkaido University, Sapporo 060-8628, Hokkaido, Japan; wuyuwei@eis.hokudai.ac.jp (Y.W.); beeth_karajan@eis.hokudai.ac.jp (T.Y.); rusaruka_mareusu@eis.hokudai.ac.jp (S.O.)

[2] Faculty of Engineering, Hokkaido University, Sapporo 060-8628, Hokkaido, Japan

[3] Civil Engineering Research Institute for Cold Region, Sapporo 062-0931, Hokkaido, Japan; k.maruyama@ceri.go.jp (K.M.); ueno-c22aa@ceri.go.jp (C.U.)

\* Correspondence: t-ishika@eng.hokudai.ac.jp

**Featured Application: This article investigates the effect of wicking fabric in suppressing frost heave. The parametric analysis discusses the effectiveness of wicking fabric on different soils, different groundwater levels, and different cooling rates.**

**Abstract:** The deterioration of roads in cold regions can result in unsafe driving conditions and high maintenance costs. Frost heaving is regarded as one of the main reasons for road degradation. Generally, frost heave is caused by water migrating from the unfrozen zone to the freezing front, where it is then transformed into an ice lens. Frost heave can be reduced by removing frost-susceptible soil, raising the temperature, or removing water from the soil. Among these methods, the most economical and practical approach is to reduce the water content. Recently, an innovative geotextile known as wicking fabric (WF) has been used to drain water from unsaturated conditions and minimize frost heaving. The objective of this study was to evaluate the inhibition effects of WF on frost heave under different experimental conditions in the freezing process. In this study, a thermo-hydro-mechanical (THM) coupled numerical model is proposed to simulate the freezing process of subgrade soil with WF. The evaporation model is used to simply describe the water absorption characteristics of WF. The numerical model was validated by comparing the simulation results with the experimental results of the wicking fabric model (WWF) and the non-wicking fabric model (NWF). Additionally, parametric analysis was conducted to examine the effectiveness of WF in reducing frost heave under various experimental conditions. As a result, the freezing process of soil installed with WF was accurately simulated by the proposed model. WF showed inhibition effects on frost heave under various experimental conditions. The results indicate the following: (1) Compared to Touryo soil (a high frost-susceptible clay-sand soil), WF inhibited frost heave more effectively in Tomakomai soil (a medium frost-susceptible lean clay), while the inhibition effect of WF in Fujinomori soil (a medium frost-susceptible lean clay) was limited. (2) WF has a more significant frost heave inhibition effect at a slower cooling rate in the freezing process. (3) The further the WF is installed from the groundwater level (GWL), the greater its impact on inhibiting frost heave.

**Keywords:** frost heave; water content; geotextile; wicking fabric; suction

## 1. Introduction

In cold regions, the cold weather generally causes frost heaving, which can damage roads, railroads, buildings, and airports [1]. In cases of severely damaged pavement, vehicles must reduce their speed sharply to avoid losing control. When an airfield encounters several differential frost heave problems, it may have to close parts or even all of its runways to aircraft traffic [2]. Frost heave occurs when moisture in large pore spaces freezes into ice crystals as the freezing front goes down to the ground during the freezing process. Water is constantly drawn to the freezing front by capillary force via frost-sensitive soils, forming ice

lenses. For frost heave to occur, three conditions must be met: freezing temperature, frost-susceptible soils, and the presence of a continuous source of water [3]. Removing any of the three factors above will eliminate or dramatically reduce frost heave. Typical measures include replacing frost-susceptible soils with non-susceptible soils, installing insulation layers to reduce frost penetration, increasing the overburden pressure, and installing a capillary barrier to stop the water flow to reduce surface water accumulation [4–6]. Given the high cost of transporting materials to remote sites and the long distance, the most practical method for reducing frost heave is reducing the water content in the pavement structure [7]. Recently, geotextiles and geocomposites have been used as capillary barriers in pavements to reduce water absorption and frost heaving. However, capillary barriers only obstruct the upward flow of capillary water. The excess water could accumulate beneath the capillary barriers, ultimately reducing the pavement stiffness.

A new type of woven geotextile with wicking ability was recently developed to minimize upward water migration and drain excess water effectively [8]. As shown in Figure 1, when WF is installed in the subgrade, the air humidity above the roadway is typically dry (less than 50% relative humidity), the soil, as well as the geotextile, is wet (close to 100% relative humidity), and a large suction gradient would form between the geotextile and soil [8]. Thus, installing WF in the subgrade helps carry both gravitational and capillary water to the face of the road slopes and eventually evaporate to the ambient atmosphere. Specifically, upwardly migrating moisture that reaches the WF can be laterally diverted to the pavement shoulders [9].

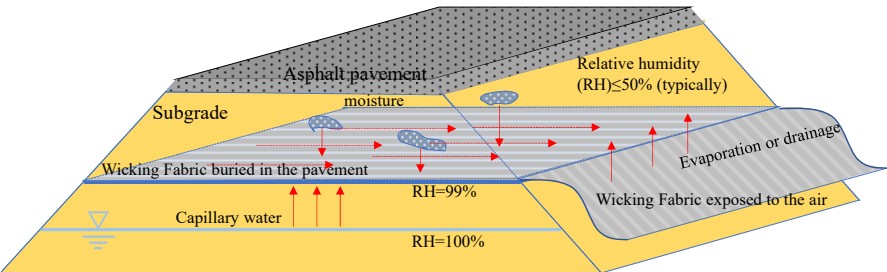

**Figure 1.** Mechanisms of water removal by WF in a pavement system.

Researchers have conducted several successful laboratory tests and field applications to investigate the drainage efficiency of wicking fabrics in recent years. For instance, Zhang and Presler [8] installed two layers of WF at the Beaver Slide area of the Dalton Highway and successfully prevented frost heave and subsequent thaw weakening; Lin et al. [7] found that the WF installed five years ago in that project conducted by Zhang and Presler [10] continued functioning well. According to Currey [11], WF has also been successfully applied to soft subgrades, saving USD 2.5 million in the initial construction phase. In addition, it has also been used to address differential settlement-induced pavement deterioration by Delgado [12]. Han et al. [13] analyzed the microstructure of WF and presented an explanation for the water absorption working mechanism of this material. An experiment conducted by Guo et al. [14] indicated that the capillary of the WF could effectively drain water from the soil column up to a considerable distance through the WF. Lin et al. [15] performed a comprehensive laboratory test to characterize the property of the WF and soil–WF interactions, and the results showed that the WF can effectively drain water until the suction reaches the inner-yarn air entry value. Furthermore, several researchers have attempted to simulate the drainage process of WF with numerical tools. For instance, Lin et al. [15] conducted a numerical simulation to quantify the drainage ability of a soil–WF system, which demonstrated that the soil–WF system can reduce the water content of the pavement by 2% from the optimum value. Yasuoka et al. [16] performed a numerical simulation and indicated that WF could drain water and inhibit frost heave. Although several models have been developed to estimate the effectiveness of WF in draining moisture, limitations in applying the boundary condition of WF remain. Most previous numerical models do not

take into account the influence of the surrounding environment on the water absorption process of WF. In general, the water absorption process is simulated by setting a constant pressure boundary at the end of the WF. Indeed, such a simplification might still be able to reproduce the WF absorption process. Nevertheless, it has been demonstrated that the absorption effect of WF can be impacted by ambient conditions (e.g., temperature, relative humidity) [14]. It is therefore very important to consider the influence of the environment conditions when simulating the water absorption process of WF.

The primary objective of this study was to evaluate the inhibition effects of WF on frost heave under different experimental conditions in the freezing process. In this study, first, an experimental setup is presented to evaluate the inhibition effect of WF on frost heave. Next, a THM coupled numerical model is proposed, which incorporates the effects of environmental factors on the water absorption capacity of WF. The reliability of the numerical model is verified by comparing experimental and numerical results in the temperature, hydraulic, and mechanical fields in the freezing process. Furthermore, the proposed model is also used to simulate the frost heave amounts in the WWF and NWF models under various conditions (groundwater level, soil type, and cooling rate). These findings are used to evaluate how different factors affect the ability of WF to inhibit frost heave.

## 2. Coupled THM Modeling
### 2.1. Governing Equations

In the freezing process, the soil temperature can be calculated by the heat transport equation, and the partial differential (PDE) equation for two-dimensional heat flow can be written as follows [17–19]:

$$C_v \frac{\partial T}{\partial t} = \nabla(\lambda \nabla T) + L_f \rho_i \frac{\partial \theta_i}{\partial t} \tag{1}$$

In saturated–unsaturated soils, the water and energy fluxes through the boundary lead to moisture transformation and phase change; this will cause variations in the water content and internal energy. Richard's equation can be used to describe the water movement in a variably saturated porous medium, and in this equation, the liquid water flow in frozen soil is analogous to unfrozen soil and can still be described by Darcy's law. In this study, a mixed form of Richard's equation was adopted to ensure that mass conservation was maintained [20], which can be written as Equation (2) [21,22],

$$\frac{\partial \theta_u}{\partial t} + \frac{\rho_i}{\rho_u} \frac{\partial \theta_i}{\partial t} = \nabla(K_r \nabla h + K\boldsymbol{i}) - SI \tag{2}$$

*SI* is the sink term. As the conduction effect on the heat transfer and water migration is limited, it is generally neglected in most studies [23,24]. In this study, the sink term is applied as the actual evaporation (*AE*) caused by WF.

As for the mechanical field, Navier's equation is employed to calculate motion, strain, the displacement correlation, and constitutive relationships [25]. The general tensor format is

$$\nabla(C\nabla u) + F = \rho \ddot{u} \tag{3}$$

### 2.2. Coupled Parameters
2.2.1. Thermal Properties

The thermal conductivity of frozen and unfrozen soils can be expressed in a general way shown in the following equation [26]:

$$\lambda = \left(\lambda_{sat} - \lambda_{dry}\right)\lambda_r + \lambda_{dry} \tag{4}$$

The thermal conductivity of saturated soil and normalized thermal conductivity can be calculated by Equations (5) and (6) [26].

$$\lambda_{sat} \begin{cases} \lambda_s^{1-n} \lambda_i^{n-\theta_u} \lambda_u^{\theta_u} & T < T_f \\ \lambda_s^{1-n} \lambda_u^{\theta_u} & T \geq T_f \end{cases} \tag{5}$$

$$\lambda_{dry} = \chi \times 10^{-\eta n} \tag{6}$$

$$\lambda_r = \frac{K_0 S}{1 + (K_0 - 1)S} \tag{7}$$

The volumetric heat capacity can be calculated using a weight algorithm as

$$C_v = C_s(1 - n) + C_u \theta_u + C_i \theta_i \tag{8}$$

To reduce the non-linearity of the governing equation, the concept of apparent heat capacity can be used to merge the heat capacity with the second term of Equation (1) on the right-hand side, which means the enthalpy change is due to the phase change [27,28].

$$C_a = C_v - L_f \rho_i \frac{d\theta_i}{dT} = C_v + \frac{L_f^2 \rho_i}{gT} \frac{d\theta}{dh} \tag{9}$$

By using the generalized Clapeyron equation, the apparent volumetric heat capacity ($C_a$) can be redefined by the hydraulic capacity ($C_H$) according to Hansson's research [29]. The hydraulic capacity is defined as the derivative of the water content concerning the pressure head, which can be expressed as

$$C_H = \frac{d\theta}{dh} = \begin{cases} \dfrac{\alpha_{vg}(n_{vg} - 1)(\alpha_{vg}h)^{n_{vg}-1}}{\left((\alpha_{vg}h)^{n_{vg}} + 1\right)^{\frac{2n_{vg}-1}{n_{vg}}}} & h < h_s \\ 0 & h \geq h_s \end{cases} \tag{10}$$

where $\alpha_{vg}$ and $n_{vg}$ are the van Genuchten–Mualem fitting parameters. Therefore, Equation (9) can be rewritten as

$$C_a = C_v + \frac{L_f^2 \rho_i}{gT} C_H. \tag{11}$$

When ice generates, unfrozen water can also exist in unsaturated soil. At this time, the soil water potential remains in equilibrium with the vapor pressure over pure ice [30]. Based on the thermodynamic relationship and the Clausius–Clapeyron equation, Dall' Amico et al. [31] proposed an equation to describe the relationship between the soil matric potential and final freezing temperature in the freezing process. When $T \geq T_f$, the soil is unfrozen; when $T < T_f$, the soil is under the freezing condition.

$$T_f = T_m + \frac{gT_m}{L_f} h \tag{12}$$

where $T_m$ = 273.15 K is the nominal freezing temperature.

### 2.2.2. Hydraulic Properties

For unfrozen unsaturated soil, the hydraulic conductivity can be expressed by using the relative hydraulic conductivity, $K_{wr}$, which is the power function of the effective saturation $S_e$,

$$K_r = K_{wr} K_s = K_s S_e^{\frac{1}{2}} \left[ 1 - \left( 1 - S_e^{\frac{\lambda_{vg}}{\lambda_{vg}-1}} \right)^{\frac{\lambda_{vg}-1}{\lambda_{vg}}} \right]^2 \tag{13}$$

Using the Mualem–van Genuchten [32] model, $S_e$ can be expressed as

$$S_e = \frac{\theta - \theta_r}{\theta_s - \theta_r} = \left[1 + \left(\alpha_{vg}h\right)^{\lambda_{vg}}\right]^{\frac{1-\lambda_{vg}}{\lambda_{vg}}} \tag{14}$$

In frozen soil, the permeability decreases as the ice saturation increases. Considering the ice content, Jame and Norum [33] adopted an impedance factor approach to describe the permeability in the freezing process, which can be written as

$$K_r = K_s I \tag{15}$$

Taylor and Luthin [34] compared the simulation result with the data of Jame [35], proposing a relationship between the volumetric ice content and the impedance factor, which showed acceptable agreement. The impedance factor $I$ was adopted according to the study of Taylor and Luthin [34],

$$I = 10^{-E_i \theta_i} \tag{16}$$

Based on Shoop and Bigl [36], $E_i$ represents an empirical parameter related to the saturated hydraulic conductivity and can be expressed as follows:

$$E_i = \frac{5}{4}(K_s - 3)^2 + 6 \tag{17}$$

Similarly, for unsaturated frozen soil, the permeability can be expressed as

$$K_r = IK_s S_e^{\frac{1}{2}} \left[1 - \left(1 - S_e^{\frac{\lambda_{vg}}{\lambda_{vg}-1}}\right)^{\frac{\lambda_{vg}-1}{\lambda_{vg}}}\right]^2 \tag{18}$$

Suppose $h_s$ is the saturated matric potential, that is, $h_s = 0$. Equations (13) and (18) describe the unsaturated soil hydraulic conductivity in the freezing process. Thus, the formulation of the hydraulic conductivity $K_r$ under freezing conditions, for saturated and unsaturated soils, becomes

$$K_r = \begin{cases} h \geq h_s & \begin{cases} k_s & T \geq T_f \\ Ik_s & T < T_f \end{cases} \\ h < h_s & \begin{cases} k_s S_e^{\frac{1}{2}} \left(1 - \left(1 - (S_e)^{\frac{n_{vg}}{n_{vg}-1}}\right)^{\frac{n_{vg}-1}{n_{vg}}}\right)^2 & T \geq T_f \\ Ik_s S_e^{\frac{1}{2}} \left(1 - \left(1 - (S_e)^{\frac{n_{vg}}{n_{vg}-1}}\right)^{\frac{n_{vg}-1}{n_{vg}}}\right)^2 & T < T_f \end{cases} \end{cases} \tag{19}$$

### 2.2.3. SWCC and SFCC

The soil–water characteristic curve (SWCC) represents the relationship between suction and the volumetric water content in the unfrozen status. When the soil is saturated, the unfrozen volumetric water content can be expressed as the saturated volumetric water content. Meanwhile, for the unsaturated condition, the unsaturated soil hydraulic properties may be used to describe the volumetric water content. In this study, the van Genuchten–Mualem equation [32] with independent $\alpha_{vg}$ and $n_{vg}$ parameters was used.

$$\theta_u = \begin{cases} (\theta_s - \theta_r)\left(\dfrac{1}{\left(|h|\alpha_{vg}\right)^{n_{vg}} + 1}\right)^{\frac{n_{vg}-1}{n_{vg}}} + \theta_r & h < h_s \\ \theta_s & h \geq h_s \end{cases} \tag{20}$$

Meanwhile, a similar soil freezing characteristic curve (SFCC) was used to describe the relationship between the temperature and suction in the freezing process. Assuming that the ice pressure is taken as zero, Hansson et al. [29] put forward a generalized Clapeyron equation based on thermal dynamic equilibrium theory. The Clapeyron equation can be used to convert the sub-freezing temperature to suction. The SFCC can be derived from the SWCC by relating suction to the sub-freezing temperature based on this theory. When the temperature is under the sub-freezing point, the equation to obtain the soil matric potential is written as

$$h = \frac{L_f}{g} In \frac{T}{T_f} \tag{21}$$

### 2.3. Actual Evaporation of WF

The wetting process describes the movement of absorbed water in the geotextile. Geotextiles can remove water through gravity drainage and evaporation during a wet period. In saturated soils, gravity drainage occurs if sufficient water is supplied and the rate of water flow onto an exposed geotextile is greater than the evaporation rate. WF and conventional geotextiles can both provide gravity drainage. Besides serving as a gravity drainage device, WF can remove water in unsaturated soils by sucking it into the fibers. The WF water migration process is illustrated in Figure 2a, which is modified from Guo [37]. The water is transported to an exposed portion of the WF and evaporates into the air. In this study, the water removed by WF is described as an evaporating process, as illustrated in Figure 2a. The model test simulation was simplified by using the following assumption for the evaporation process of WF: water evaporation from WF exposed to air (*X*) equals the water absorbed from WF buried in the soil (*Y*), as shown in Figure 2b. Accordingly, the numerical model calculated the evaporation of WF exposed to air and applied it to the WF buried in the soil as a line source. However, this simplification is limited. As the model assumes that the rate of evaporation from the WF is the same as the rate of water removal from the soil, this assumption only holds true when the area of WF exposed to air (*La*) is not less than the area buried in the soil (*Ls*). Therefore, when the area of the WF exposed to air is smaller than that buried in the soil, the evaporation amount calculated by this method will be higher than reality. Accordingly, the WF appears to be more effective at suppressing frost heave in the calculation than it is.

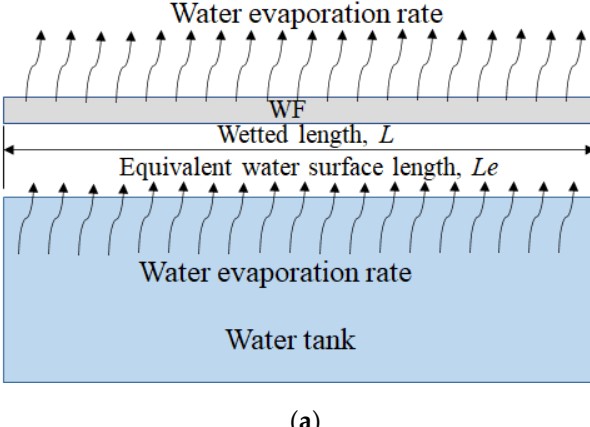

(a)

**Figure 2.** *Cont.*

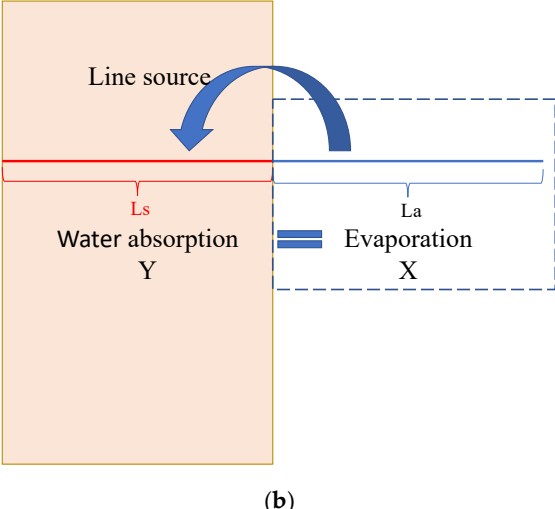

**(b)**

**Figure 2.** Schematic of water transport and evaporation in wicking geotextile. (**a**) Water transport and evaporation in wicking geotextile 5. (**b**) Simplified schematic of the evaporation model.

Referring to Fredlund et al. [38], the actual evaporation rate can be expressed in the form of the thermodynamic equilibrium relationship between the relative humidity and total suction,

$$AE = PE \times \exp^{\left(\frac{-\psi g w_v}{\zeta(1-RH\gamma w)RT}\right)} \tag{22}$$

The widely used equation to calculate the potential evaporation rate was proposed by Penman [39], which is written as

$$PE = \frac{\Gamma Q_n + \eta_e E_a}{\Gamma + \eta_e} \tag{23}$$

where $E_a$ can be expressed as

$$E_a = 3.5(1 + 0.146U_a)(e_{a0} - e_a) \tag{24}$$

The heat budget is caused by the net radiant energy available at the surface. The heat budget and the wind speed are neglected in the freezing process, so Equation (23) can be simplified as

$$PE = \frac{\eta_e 3.5(e_{a0} - e_a)}{\Gamma + \eta_e} \tag{25}$$

Tetens [40] estimated $\Gamma$ based on the air temperature as follows:

$$\Gamma = \frac{4098e_{a0}}{(273.15 + T_a)^2} \tag{26}$$

and

$$e_{a0} = \frac{e^0(T_{amax}) + e^0(T_{amin})}{2} \tag{27}$$

$$e_a = \frac{e^0(T_{amax})\frac{RH_{amax}}{100} + e^0(T_{amin})\frac{RH_{amin}}{100}}{2} \tag{28}$$

$$e^0(T_a) = 0.6108 \exp\left(\frac{17.27T_a}{T_a + 273.15}\right) \tag{29}$$

This study simulates the absorption of water by WF as an evaporation process and models this process numerically. In contrast to the evaporation process occurring under

saturated conditions with sufficient water supply, the unsaturated soil evaporation model is coupled with the evaporation process of the WF. In addition, this study presents a numerical simulation of the WF evaporation process under unsaturated conditions. The model includes both the evaporation and freezing processes. Thus, the proposed model can be used to simulate the water uptake of WF in unsaturated soil during the freezing process.

Table 1 contains all abbreviations or symbols applied to variables and parameters, and Table 2 contains the fixed parameters which are employed in the model.

**Table 1.** List of parameters.

| Abbreviation/Symbol | Parameter/Variable | Units |
|:---:|:---:|:---:|
| $C_v$ | volumetric heat capacity of the soil mixture | J/m$^3$·K |
| $C_s$ | heat capacity of the soil particles | J/m$^3$·K |
| $C_u$ | heat capacity of the | J/m$^3$·K |
| $C_i$ | heat capacity of the | J/m$^3$·K |
| $C_a$ | apparent volumetric heat capacity | J/m$^3$·K |
| $C_H$ | hydraulic capacity | J/m$^3$·K |
| $\lambda_{sat}$ | thermal conductivity of saturated soil | W/(m·K) |
| $\lambda_{dry}$ | thermal conductivity of dry soil | W/(m·K) |
| $\lambda_s$ | thermal conductivity of the soil | W/(m·K) |
| $\lambda_r$ | normalized thermal conductivity | W/(m·K) |
| $\lambda_i$ | thermal conductivity of ice | W/(m·K) |
| $\lambda_u$ | thermal conductivity of unfrozen water | W/(m·K) |
| $\lambda$ | thermal conductivity of the soil mixture | W/(m·K) |
| $n$ | porosity of the soil | 1 |
| $T_f$ | freezing temperature of the soil | K |
| $\chi$ | material parameters accounting for the particle shape effect | 1 |
| $\eta$ | material parameters accounting for the particle shape effect | 1 |
| $K_0$ | an empirical parameter used to account for the different soil types in the unfrozen and frozen states | 1 |
| $T$ | temperature in the soil mixture | K |
| $\theta_u$ | volumetric unfrozen water content | 1 |
| $\theta_i$ | volumetric ice content | 1 |
| $S$ | degree of saturation | 1 |
| $S_e$ | effective saturation | 1 |
| $K_s$ | saturated water hydraulic conductivity | ms$^{-1}$ |
| $h_s$ | saturated matric potential | ms$^{-1}$ |
| $K_r$ | hydraulic conductivity of the soil | ms$^{-1}$ |
| $h$ | total hydraulic head | m |
| $\boldsymbol{i}$ | unit vector along the direction of gravity | 1 |
| $C$ | fourth-order tensor of material stiffness | 1 |
| $u$ | displacement vector | 1 |
| $F$ | body force vector | 1 |
| $\psi$ | matric suction | kPa |
| $RH$ | relative humidity of the overlaying air | 1 |
| $AE$ | actual evaporation | mmday$^{-1}$ |
| $PE$ | potential evaporation | mmday$^{-1}$ |
| $\Gamma$ | the slope of the saturation vapor pressure versus the temperature curve at the mean temperature of the air | 1 |
| $Q_n$ | heat budget | 1 |
| $E_a$ | aerodynamic evaporative term | 1 |
| $e_{a0}$ | saturation vapor pressure of the mean air temperature | kPa |
| $e_a$ | actual vapor pressure of the air | kPa |
| $U_a$ | wind speed | ms$^{-1}$ |
| $e^0(T_a)$ | saturation vapor pressure of the air at the air temperature $T_a$ | kPa |
| $T_{amax}$ | maximum temperature of the air | K |
| $T_{amin}$ | minimum temperature of the air | K |
| $RH_{amax}$ | maximum relative humidity of the air | 1 |
| $RH_{amin}$ | minimum relative humidity of the air | 1 |

**Table 2.** Physical constants and parameter values used in the model.

| Parameter | | Value | Units |
|---|---|---|---|
| $L_f$ | latent heat of fusion | 334,000 | J/kg$^{-1}$ |
| $\rho_i$ | density of ice | 916 | Kgm$^{-3}$ |
| $\rho_u$ | density of water | 1000 | Kgm$^{-3}$ |
| $\zeta$ | dimensional empirical parameter | 0.7 | 1 |
| $g$ | gravity acceleration | 9.8 | m/s$^2$ |
| $w_v$ | molecular mass of water | 0.018 | kg/mol |
| $\gamma_w$ | unit mass of water | 9807 | kN/m$^3$ |
| $R$ | universal gas constant | 8314 | J/(mol·K) |
| $\eta_e$ | psychrometric constant | 66.8 | Pa/°C |

## 3. Frost Heave Test of the Soil Column

The model test is used to (1) evaluate the inhibition effect of WF on frost heave deformation, and (2) evaluate the validity of the proposed numerical model by comparing the model test result with the simulation results.

### 3.1. Material Properties and Parameters

The soils used in this study are all widely distributed in Hokkaido, Japan. Figure 3 shows the grain size distribution curves of the simulated samples. In this simulation, three typical frost-susceptible soils were used: Fujinomori soil (medium frost-susceptible loess), Touryo soil (high frost-susceptible volcanic soil), and Tomakomai soil (medium frost-susceptible volcanic soil). Fujinomori soil contains 18% clay, 78% silt, and 4% sand, while Tomakomai soil consists of 3% clay, 19% silt, and 78% sand, and Touryo soil consists of 26% clay, 21% silt, and 53% sand. According to ASTM, Touryo soil, Fujinomori soil, and Tomakomai soil are classified as clayed sand (SC), lean clay (CL), and lean clay (CL), respectively. The soil input parameters used in the numerical simulation are listed in Table 3.

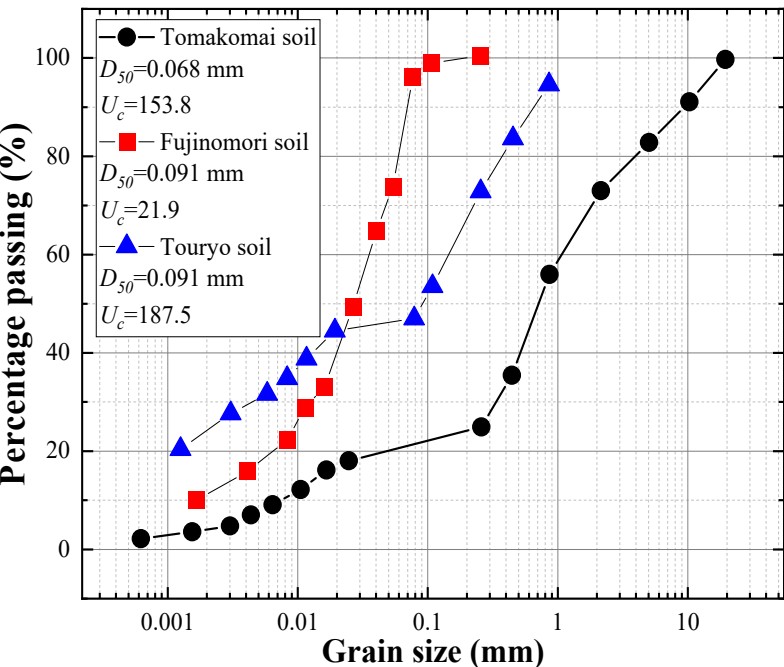

**Figure 3.** Grain size distribution curve of simulated soil samples.

**Table 3.** List of soil parameters.

| Abbreviation/ Symbol | Parameter | Value | | | Unit |
|---|---|---|---|---|---|
| | | **Touryo Soil** | **Fujinomori Soil** | **Tomakomai Soil** | |
| $C_s$ | Volumetric heat capacity of the soil particles | $1.8 \times 10^6$ | $1.3 \times 10^6$ | $8.59 \times 10^5$ | J/m$^3$·K |
| $\lambda_s$ | Thermal conductivity of the soil mixture | 1.61 | 0.83 | 1.61 | W/(m·K) |
| $\chi$ | Material parameters accounting for the particle shape effect | 0.75 | 0.75 | 0.75 | W/(m·K) |
| $\eta$ | Material parameters accounting for the particle shape effect | 1.2 | 1.2 | 1.2 | 1 |
| $\rho_d$ | Dry density of soil particles | 1400 | 1460 | 1200 | Kgm$^{-3}$ |
| $n$ | Porosity | 0.45 | 0.455 | 0.55 | 1 |
| $T_m$ | Final freezing temperature at atmospheric pressure | 272.95 | 272.90 | 273.05 | K |
| $\alpha_{vg}$ | Van Genuchten–Mualem fitting parameter | 93.2 | 1.904 | 25.02 | MPa$^{-1}$ |
| $\lambda_{vg}$ | Van Genuchten–Muale fitting parameter | 1.596 | 1.865 | 1.54 | 1 |
| $S_s$ | Saturated degree of saturation | 96.7 | 100 | 95.1 | % |
| $S_r$ | Residual degree of saturation | 37.8 | 18.5 | 33.5 | % |
| $k_s$ | Saturated water hydraulic conductivity | $1 \times 10^{-8}$ | $5 \times 10^{-10}$ | $9.16 \times 10^{-9}$ | ms$^{-1}$ |
| $\alpha_{Tu}$ | Thermal expansion coefficient | $1.2 \times 10^{-5}$ | $1.2 \times 10^{-5}$ | $1.2 \times 10^{-6}$ | K$^{-1}$ |
| $E$ | Young's modulus of soil | 40 | 12.5 | 8.5 | MPa |
| $H$ | Modulus related to matric potential | 7653 | 7653 | 7653 | m |
| $\upsilon$ | Poisson's ratio | 0.4 | 0.33 | 0.4 | 1 |

The dry density and porosity of each soil sample are determined from the average of the results from each test group. Based on the recommendations of Guymon et al. [41], the thermal conductivity and volumetric heat capacity of soil particles were determined according to the results of water retention tests. The Mualem–van Genuchten model parameters used in Equations (13) and (14) were obtained through parameter fitting to the water retention test [42]. Based on the permeability test results [43,44], the saturated hydraulic conductivity was determined for each soil. It should be noted that thermal expansion is considered to have only a marginal effect on the frost behavior of soil. Thus, the thermal expansion coefficient for water was set to zero. Other parameters were given by previous studies [45–48].

*3.2. Experimental Setup*

Figure 4a is a schematic diagram of a frost heave test device installing WF. Two thermostatic baths, which circulate an anti-freezing liquid at a specific temperature, are used to control the temperature of the upper cooling plate and lower cooling plate independently. The porous metal plate at the bottom of the device is connected to a water tank, to maintain a constant groundwater level (GWL) for the device. At the height of 150 mm from the pedestal, on one side of the model, a sliding opening of 100 mm in width and 0–2.5 mm in height is installed to place the geotextile. The specimen is installed with five moisture sensors (ECH2O soil moisture sensor), five thermal sensors (T-type thermocouples), and two suction sensors (tensiometers) to measure temperature, saturation, and suction at the observation point in the freezing process. The soil surface temperature is measured using a platinum resistance temperature detector. Moreover, the displacement transducer is used to measure the specimen frost heave. The applied load represents the forces acting on the subgrade in the actual pavement structure.

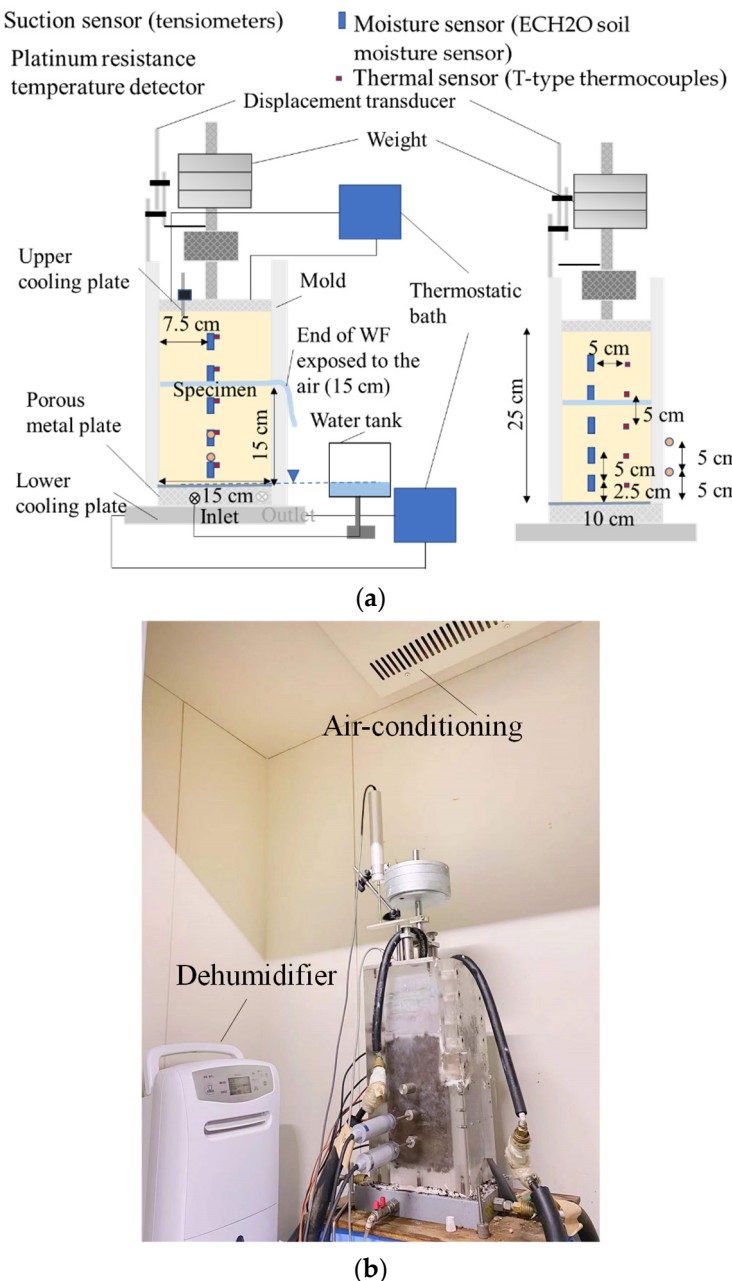

**Figure 4.** Frost heave model test. (**a**) Schematic of the test apparatus. (**b**) Picture of the test setup.

### 3.3. Testing Method

#### 3.3.1. Soil Column Preparation

The selected soil sample was obtained from Tomakomai, Hokkaido, where the soil is usually used as a subgrade layer. In addition, a density test (JIS A 1202:2009) [49], a sieve analysis (JIS A 1204:2009) [50], and a compaction test (JIS A 1210:2009) [51] were conducted. The density ($\rho_s$) and the maximum dry density ($\rho_{dmax}$) of the soil were 2.64 g/cm$^3$ and 1.27 g/cm$^3$, respectively. The uniformity coefficient (*Uc*) was 153.8. This soil can be classified as SFG (sand fine particle and gravel) according to JGS 0051 [52]. The optimum water content of the Tomakomai soil is 32.0%. After the in situ soil was dried, it was sprayed with water until the water content reached 90% and sealed with plastic bags to prevent the evaporation of moisture and obtain a uniform water content distribution.

The test apparatus is shown in Figure 4a. The size of soil columns for both the NWF model and the WWF model was 250 mm in height, 150 mm in width, and 100 mm in depth. This geometry size was selected to ensure the sensor readings are precise, and to ensure

that the model is able to observe the drainage scope of the buried WF. In the NWF model experiment, the wet soil samples were placed into the mold in 5 layers and compacted by layers from bottom to top. Each layer was compacted using a 2.5 kg rammer in order to achieve compaction of 95.0% ($\rho_d$ = 1.21 g/cm$^3$) and saturation of 90%. In order to shorten the saturation time before freezing, the saturation of the soil samples was prepared to 90%. To prevent stratification of the soil, the surface of layers 1–4 was scraped after compaction. Before filling in the next layer of soil samples, a T-type thermocouple and a moisture meter were placed. The first layer of soil samples was 2.5 cm deep, and the second to fifth layers were each 5 cm thick. On the other hand, in the WWF model experiment, a dry WF (100 mm wide by 300 mm long) was laid into the soil after the soil sample was filled to a depth of 150 mm, and its end was exposed from the preset slide at a length of 150 mm. The grease was used to fill the space between the mold and the WF during the installation of the WF. Afterward, the 4th and 5th layers of the soil were placed on the top.

### 3.3.2. Test Procedures

Figure 4b shows the test setup of the frost heave model test. An overburden pressure of 10 kPa was applied at the top of the specimen in accordance with the pavement thickness in the field test. The frost heave model test of the soil column was carried out as follows. First, the initial condition of a test specimen was set with the saturation process followed by the drainage process, under a room temperature of 9 °C and a room humidity of 50%. In the saturation process, the GWL was set to 0.25 m by adjusting the height of the water tank connected to the soil specimen, and the water inlet connected to the porous metal plate was opened, allowing water to flow into the soil specimen. When the temperature and water content of the soil specimen no longer fluctuated, the GWL was adjusted to the bottom of the soil specimen (0 m) to drain the water out. This status is considered as the initial condition of the soil specimen before the freezing process. It should be noted that the initial water content distribution for the WWF model is different from that for the NWF model. In the WWF model, the initial water content decreases more than the NWF model due to the drainage of the WF.

Next, once it was confirmed that the measured temperature and volumetric water content of the specimen were stable, the freezing process was conducted under a room temperature of 0 °C and a room humidity of 50%. At first, the temperatures of the upper and lower cooling plates were set at 0.5 °C using a thermostatic bath. This temperature was maintained until the soil specimen achieved a steady state. After that, the temperature of the upper cooling plate was rapidly dropped to −10 °C (thermal shock) while keeping the temperature of the lower cooling plate constant (JGS 0171). By generating latent heat, the supercooled state was avoided. Following the thermal shock, the temperature of the upper cooling plate was returned to −1.0 °C, while the temperature of the lower cooling plate was maintained at 0.5 °C. To maintain the freezing rate in the range of 1 mm/h to 2 mm/h (JGS 0172), the temperature of the upper cooling plate was reduced from −1.0 °C in a constant gradient ($U$ = 0.1 °C/h) for 100 h. During the freezing process, the water supply valve was kept open so that the water could migrate from the water tank to the soil specimen via the inlet. In addition, the mold had an opening that enabled one end of the WF to be exposed to air, where the moisture absorbed by the WF evaporated or drained out.

## 4. Numerical Simulations with Coupled THM Analysis

The assumption of two-dimensional plane strain was made, and the numerical simulations were performed under various test conditions, including WWF and NWF models, with different frost-sensitive soils, cooling rates ($U$), and GWLs ($l$). Moreover, assuming that the bottom of the soil sample is the origin and upwards is positive, $l$ is the distance from the origin to the upper surface of the saturated zone in the soil. The size of the sample, the boundary conditions, and the mesh of the two-dimensional model using nine-node quadrilateral elements are presented in Figure 5. It is noted that the preliminary examinations of the mesh sensitivity showed no significant difference in the simulation results

by the mesh density when using the mesh shown in Figure 4 and a mesh finer than that shown in Figure 5. For this reason, the mesh in Figure 4 was used in this study to ensure the convergence of the results while maintaining the accuracy of the calculations and taking up fewer computational resources. However, the water content near the WF varies significantly during the freezing process. Thus, in the WWF model, a refined grid was adopted near the WF. In this study, the observation points were placed at the same locations as the measured sensors in the model test, for the consistency between the numerical simulation and the test results. There are five observation points located at the height ($d$) of 25 mm, 75 mm, 125 mm, 175 mm, and 225 mm away from the origin. The dimensions of the soil column are 250 mm × 150 mm (H × W), which are similar to the actual model test. It is noted that the bottom of the soil is set at a baseline (0 mm), and the value ($d$) is considered upward as positive.

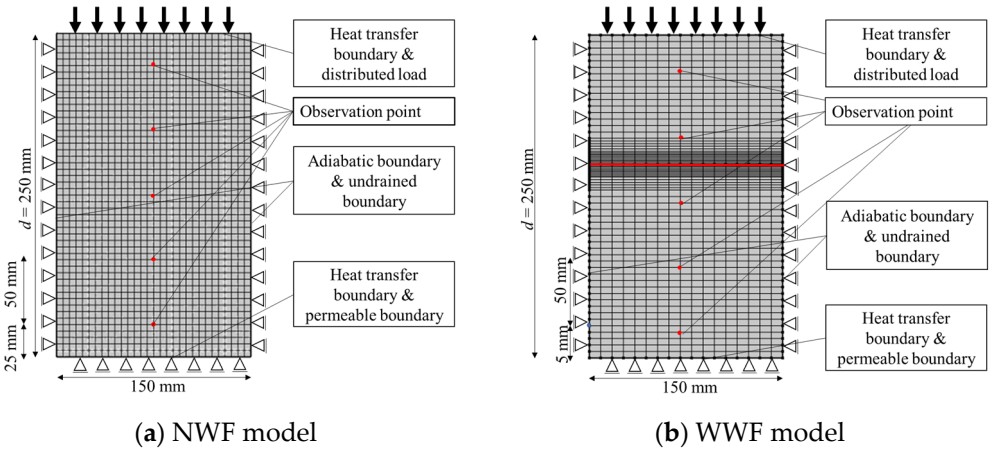

**Figure 5.** Numerical model and boundary conditions for frost heave tests.

The initial boundary conditions of the NWF model were set as follows. The base surface was fixed vertically, and its lateral boundaries were fixed horizontally in the mechanical field. The hydrostatic pressures were applied to the base surface to maintain a constant GWL. Impermeable boundaries were applied to the lateral sides and top surface in the hydraulic field. The adiabatic boundary condition was applied on the two lateral sides, while on the top and bottom surfaces, the constant thermal boundary condition was applied. Except for the suction boundary caused by the WF, the boundary conditions for the WWF model were the same as those for the NWF model. As shown in Figure 4b, the evaporation model (Equation (22)), working as a line source, was applied to the WF to simulate the drainage process caused by the WF. The water absorption process occurs in this scenario at the WF buried within the specimen.

## 5. Results and Discussions

### 5.1. Reliability of the Proposed Model

The numerical results were compared with the experimental data to validate the proposed model. All these experimental and numerical results were obtained from their respective single tests. Figure 6 shows comparisons of the temperature distribution for the soil columns at different observation points in the numerical and experimental studies. The simulation results for both the WWF and NWF models are consistent with the experimental results at each observation point. Additionally, the comparison of the results between the WWF model and the NWF model reveals that these two models do not show significant differences in the temperature distribution and variations during the freezing process. The results demonstrate that the proposed model can accurately simulate temperature changes in the WWF and NWF models during the freezing process.

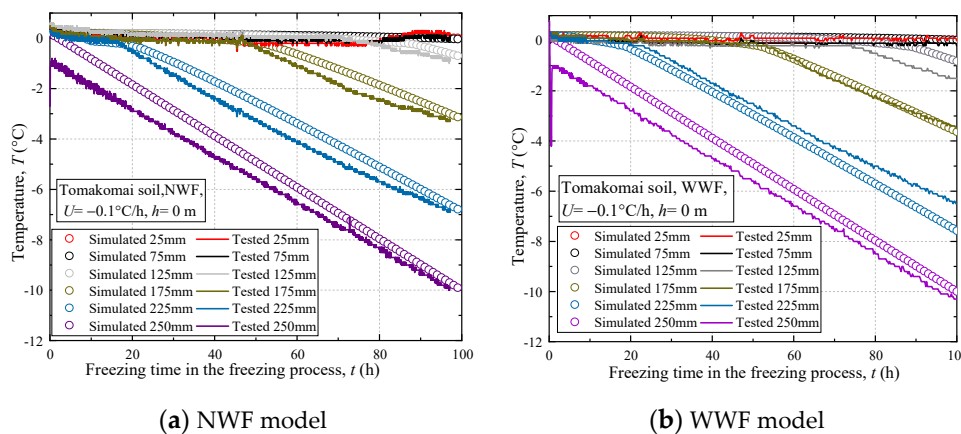

(**a**) NWF model        (**b**) WWF model

**Figure 6.** Comparison of the simulated and tested temperature development at the different observation points.

Figure 7 presents the volumetric water content variation in the freezing process at the different observation points of the simulation results and experimental results. In the WWF model, the simulation results for the water content at 125 mm differ significantly from the experimental results. The moisture sensor at this site was found to be broken when checking the experimental equipment after the test. However, the numerical calculations were consistent with the experimental results for other observation points. The results indicate that the proposed model is capable of predicting the development and distribution of the water content in the freezing process.

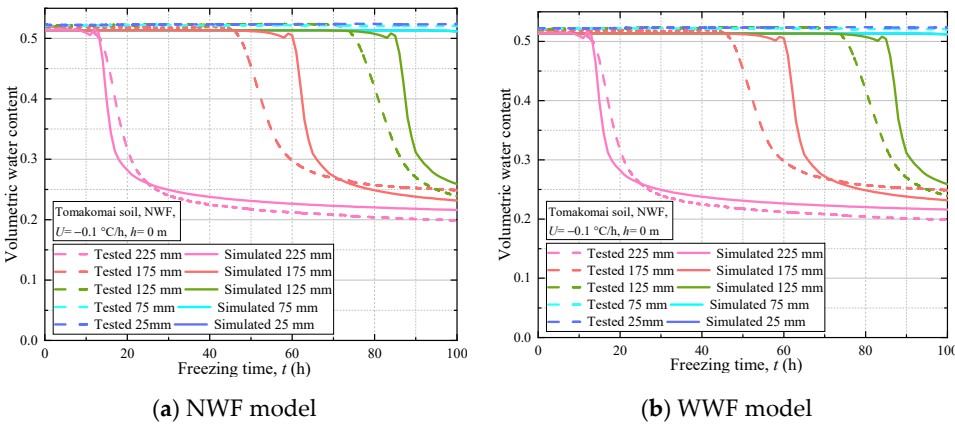

(**a**) NWF model        (**b**) WWF model

**Figure 7.** Comparison of the simulated and tested volumetric water content at the different observation points.

In both the WWF and NWF models, the volumetric water content decreased rapidly once freezing began; the closer the observation point is to the cold end, the more significant the reduction in the volumetric water content. Furthermore, Table 4 presents the freezing rate ($U_0$), frost heave rate ($U_h$), frost heave ratio ($\zeta$), and frost penetration depth ($l_0$) in the model tests and numerical simulations. The freezing rate ($U_0$), frost heave rate ($U_h$), frost heave ratio ($\zeta$), and frost penetration depth ($l_0$) stand for the rate at which the freezing front advances into the unfrozen soil, the amount of frost heave per unit time, the increase in the volume of soil due to freezing expressed as a percentage of the volume before freezing, and the depth to which the groundwater in the soil is expected to freeze (JGS 0171-2009, JGS 0172-2009) [53,54]. The freezing rate is determined by the freezing time and the frost penetration depth. Assuming the bottom surface as the origin and *d* as the distance from the origin, with upwards as positive, freezing begins at the top surface (*d* = 250 mm). The freezing rate was calculated by dividing the distance between *d* = 225 mm and *d* = 125 mm by the time differential between the freezing front developed at the two points. Multiplying

the freezing rate by the freezing time (100 h) returns the frost penetration depth. Both in the simulations and model tests, the WWF model had a deeper frost penetration depth than the NWF model. As a result of the lower water content due to the WF, the heat capacity of the soil mixture in the WWF model was smaller than that in the NWF model, and the thermal conductivity was higher. Therefore, under the same freezing conditions, the WWF model took less time to freeze to the same height as the NWF model. Comparing at the same observation point and the same freezing time, the volumetric water content in the WWF model was much smaller than that in the NWF model. Additionally, the water content in both the WWF and NWF models decreased significantly with increasing freezing time. Compared to the NWF model, the WWF model showed a more significant decrease in the water content over time. This phenomenon occurs because the water in the WWF model is sucked out by the WF.

**Table 4.** Frost heave test and simulation results.

|  | WWF | | NWF | |
|---|---|---|---|---|
|  | **Simulated** | **Tested** | **Simulated** | **Tested** |
| $U_0$ (mm/h) | 1.724 | 1.882 | 1.587 | 1.562 |
| $U_h$ (mm/h) | 0.157 | 0.156 | 0.276 | 0.295 |
| $l_0$ (mm) | 172.4 | 188.2 | 158.7 | 156.2 |
| $\zeta$ (%) | 6.28 | 6.26 | 11.04 | 11.79 |

A comparison of the measured frost heave strain during the frost heave test with the simulated results is shown in Figure 8. It appears that the simulation results agree well with the experiment results. It should be noted that the WWF model produced a significantly lower frost heave rate and ratio than the NWF model at the same freezing time. This is due to the corresponding decrease in the water content in the WWF model since the WF has significant drainage effects [10,14]. In addition, from the comparison of the experimental data and simulation results, it is shown that the model proposed in this study accurately describes the inhibition effect of WF during freezing.

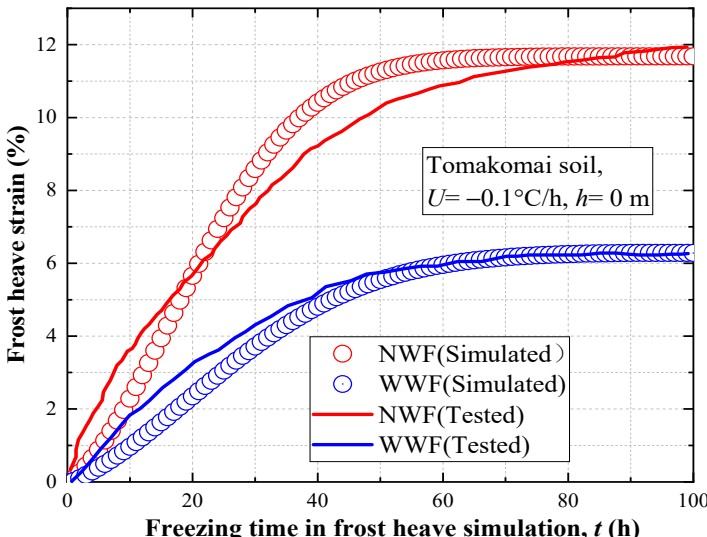

**Figure 8.** Comparison between numerical and experimental results of frost heave strain for frost heave tests of Tomakomai soil.

Figure 9 shows the change in the distribution for the degree of saturation (*S*) within the WWF and NWF models with time in the freezing process. Here, *d* represents the distance from the bottom of the soil as the origin, with upwards as positive. The saturation of the WWF and NWF models gradually decreased as the temperature of the cold end decreased.

Compared to the NWF model, the saturation around the WF was significantly reduced in the WWF model. In addition, the comparison between the WWF and NWF models revealed that the saturation in the WWF model was significantly lower than the saturation in the NWF model. Based on the saturation changes observed in the WWF model and the NWF model during the freezing process, it was found that the WF can significantly reduce the saturation during the freezing process, thereby inhibiting the development of frost heave. Furthermore, the saturation on the upper side of the WF reduced more dramatically than on the underside, and Lin et al. [15] also came to a similar conclusion in their experiment. Figure 9 illustrates that the water absorption effect of the WF in Figure 7 and the frost heave inhibition effect of the WF in Figure 8 can be rationally explained based on the simulation results with the proposed model.

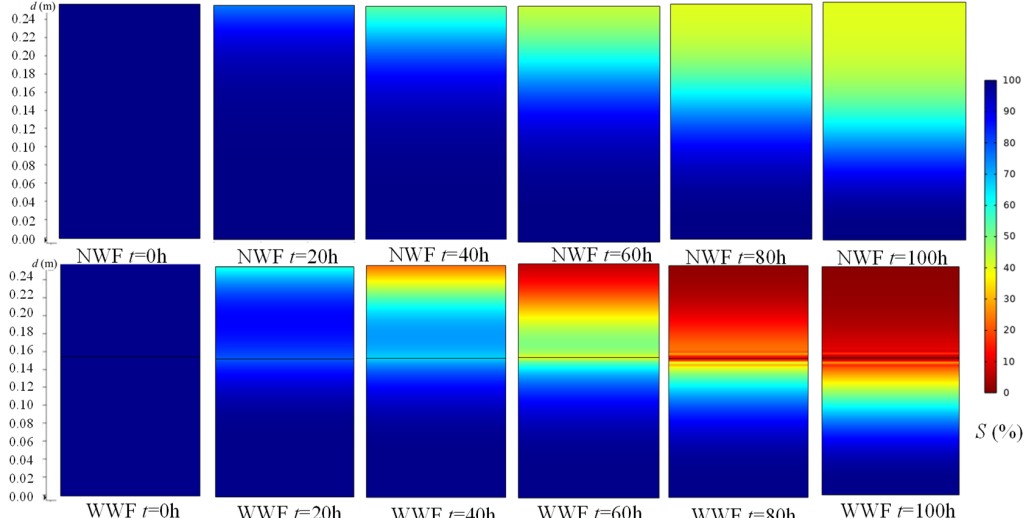

**Figure 9.** Comparison of saturation distribution for Tomakomai soil between NWF model and WWF model in the freezing process.

### 5.2. Parametric Analysis

The effects of WF on reducing frost heave may be affected by experimental conditions. As such, this study examined the factors that may influence the efficacy of WF in inhibiting frost heave. As shown in Table 5, this study conducted numerical experiments under various conditions (soil types, groundwater levels, and cooling rates). In this parametric analysis section, the WF in the WWF model simulations was set at the same height of $d = 15$ cm as in the model experiment.

### 5.2.1. Effect of Soil Types

This section investigates the frost heave inhibition effectiveness of WF in different soils using three types of frost-susceptible soils. The calculated frost heave strain of different soils in the WWF and NWF models is shown in Figure 10. The overall frost heave strain in the WWF model of different soils was lower than that in the corresponding NWF model. Comparing the frost heave inhibition efficiency of WF in different soils, it can be found that by installing WF, the frost heave strain decreased dramatically in the Touryo soil, followed by the Tomakomai soil and, lastly, the Fujinomori soil. Although WF was effective in both the Touryo soil and Tomakomai soil, it did not perform well in the Fujinomori soil. One possible explanation is that the Touryo soil has the most excellent permeability. In the NWF model, moisture can easily migrate upward to the freezing front due to the high permeability. Similarly, in the WWF model, soil with a high permeability would allow the moisture to be readily absorbed by the WWF compared to soils with lower permeability. Accordingly, the WF effect is more apparent in soils with high permeability.

**Table 5.** Numerical simulation conditions.

| Number | Soil Type | GWL $l$ (m) | $U$ (°C/h) | WF Installation |
|---|---|---|---|---|
| 1 | Touryo soil | 0 | −0.1 | WWF |
| 2 | Fujinomori soil | 0 | −0.1 | WWF |
| 3 | Tomakomai soil | 0 | −0.1 | WWF |
| 4 | Touryo soil | 0 | −0.1 | NWF |
| 5 | Fujinomori soil | 0 | −0.1 | NWF |
| 6 | Tomakomai soil | 0 | −0.1 | NWF |
| 7 | Tomakomai soil | 0.02 | −0.1 | WWF |
| 8 | Tomakomai soil | 0.04 | −0.1 | WWF |
| 9 | Tomakomai soil | 0.06 | −0.1 | WWF |
| 10 | Tomakomai soil | 0.02 | −0.1 | NWF |
| 11 | Tomakomai soil | 0.04 | −0.1 | NWF |
| 12 | Tomakomai soil | 0.06 | −0.1 | NWF |
| 13 | Tomakomai soil | 0 | −0.05 | WWF |
| 14 | Tomakomai soil | 0 | −0.2 | WWF |
| 15 | Tomakomai soil | 0 | −0.05 | NWF |
| 16 | Tomakomai soil | 0 | −0.2 | NWF |

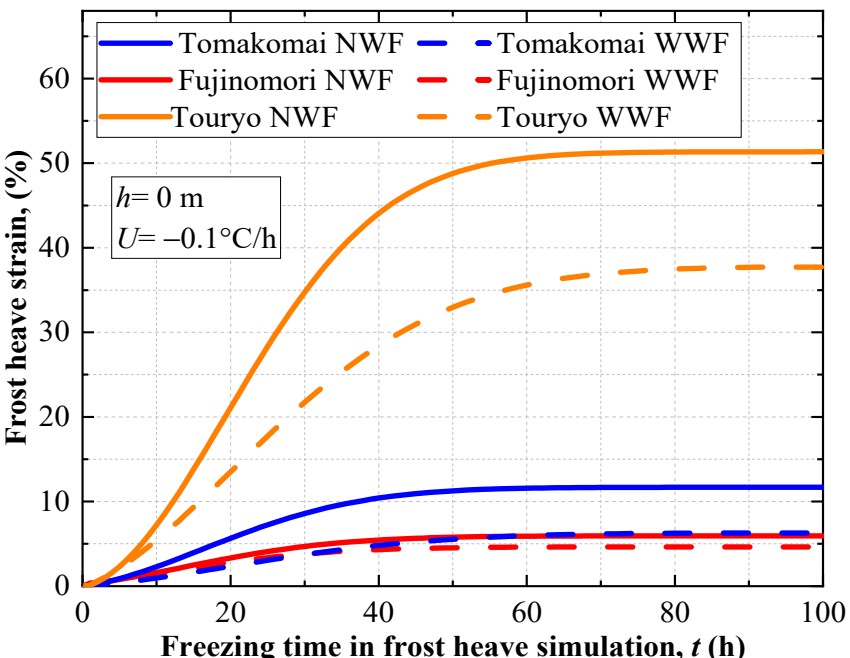

**Figure 10.** Effect of soil types on frost heave and WF inhibiting effect.

Figure 11 shows the saturation ($S$) distribution at the end of the freezing process with different types of soils in the WWF and NWF models. By comparing the saturation distributions in the WWF and NWF models for different soils, it is apparent that the saturation distributions for each of the soils in the WWF model were lower than those in the corresponding NWF model. There was a substantial decrease in saturation in the Touryo soil and Tomakomai soil by installing WF, while the reduction in soil saturation in the Fujinomori soil was relatively insignificant. The changes in the soil saturation after the installation of the WF are consistent with the observed results of frost heave inhibition. The above phenomenon indicates that WF can inhibit frost heave by reducing the water content in the soil, and the inhibition effect of WF is dependent on the permeability of the soil.

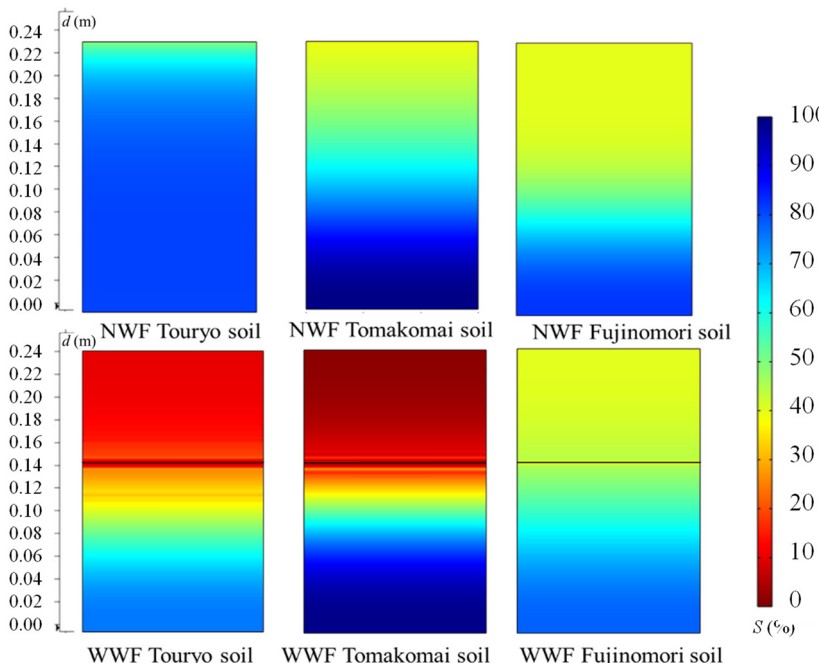

**Figure 11.** Contour plot of Effect of soil types on frost heave and WF inhibiting effect.

5.2.2. Effect of Groundwater Level

　　Frost heave and the effects of WF are also influenced by groundwater availability. Groundwater provides the mass necessary for ice formation. Based on Beskow's pioneering study [55] of frost heave, reducing the GWL reduces frost heave. Sheng et al. [56] also demonstrated that the higher the water level, the greater the amount of heave.

　　Figure 12 illustrates the simulated results of frost heave strain with time at different GWLs for a fixed installation position for the WF in the Tomakomai soil. Figure 10 illustrates how the WF was very effective in suppressing frost heave in both the Touryo and Tomakomai soils. However, the WF reduced 26.58% of the deformation in the Touryo soil, while it decreased 46.2% of the deformation in the Tomakomai soil. Therefore, in the simulation, the Tomakomai soil was found to be more effective in inhibiting frost heave in the freezing process. Hence, the Tomakomai soil was chosen as the subject of investigation in order to more clearly observe the variation in the effect of WF on frost heave inhibition in response to variances in the GWL. Moreover, the saturation (*S*) variation during the freezing process in the NWF model and WWF model is shown in Figure 13. The frost heave strain increased in both the NWF model and the WWF model as the GWL increased. It should be noted that the difference in frost heave strain between the NWF and WWF models at different GWLs was 8.448% and 6.68%, respectively, where 8.448% corresponds to a GWL at 0 mm, and 6.68% corresponds to a GWL at 60 mm. In other words, the inhibition effect of frost heave by WF decreased as the GWL increased. The following reasons may account for the impact of the GWL on frost heave strain. When the GWL increases, the soil becomes more saturated. Consequently, soil permeability increases in such circumstances. This results in a significant increase in the water flow rate and frost heave. Further, the higher the saturation, the greater the latent heat amount available in the soil, making it more difficult for the soil to freeze. In the WWF model, a higher GWL lowers the distance between the WF and GWL, therefore making it significantly more difficult to drain all the excess water immediately—in other words, reducing drain water efficiency. Additionally, a higher GWL will also lead to greater soil saturation, which in turn creates a more significant frost heave strain during the freezing process.

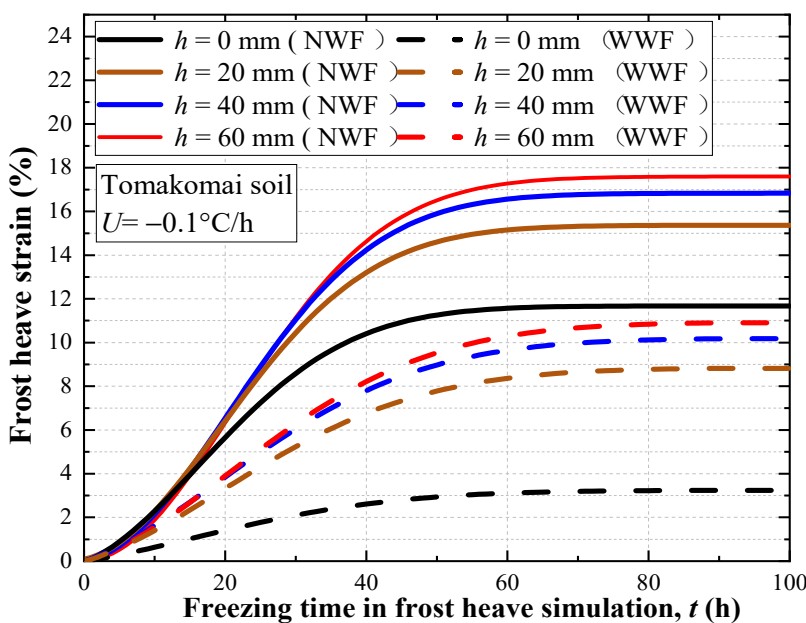

**Figure 12.** Effect of GWL on frost heave and WF inhibiting effect.

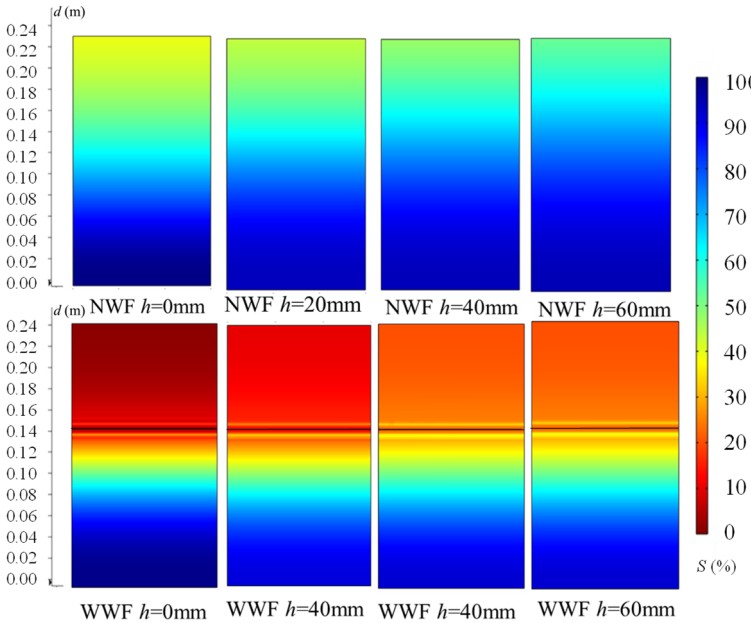

**Figure 13.** Contour plot of Effect of GWL on frost heave and WF inhibiting effect.

### 5.2.3. Effect of Cooling Rate

This section investigates the effects of the cooling rate on frost heave strain and the efficiency of drainage by the WF during the freezing process. Figure 14 illustrates the comparison between frost heave strain in the WWF and NWF models under different cooling rates. Moreover, a comparison of the saturation variations between the NWF and WWF models is presented in Figure 15 for different cooling rates during the freezing process. The WWF model shows the inhibiting effect of frost heave throughout the freezing process compared to the NWF model, regardless of the cooling rate. Moreover, according to the comparison of the simulated frost heave strain in the NWF models under different cooling rates, the frost heave strain increased as the cooling rate decreased. Based on this observation, the cooling rate has a greater impact on reducing frost heave strain during the freezing process. The slower cooling rate allows moisture to migrate upwards to the freezing front and form ice lenses, which leads to a larger amount of frost heave. A high

cooling rate can rapidly convert the water in the soil into ice, thereby reducing the soil permeability and making it more difficult for the moisture to migrate. The insufficient water supply in the high-cooling-rate circumstance will result in a limited frost heave ratio in the freezing process. Therefore, the increase in the cooling rate decreases the efficiency of the WF in inhibiting frost heave strain.

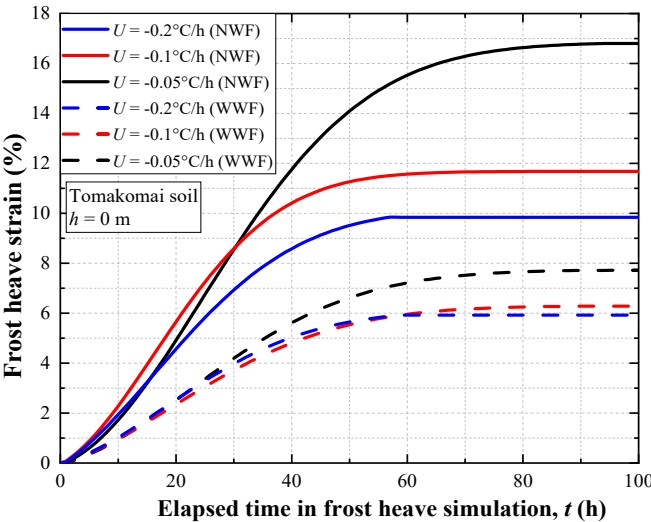

**Figure 14.** Effect of cooling rate on frost heave and WF inhibiting effect.

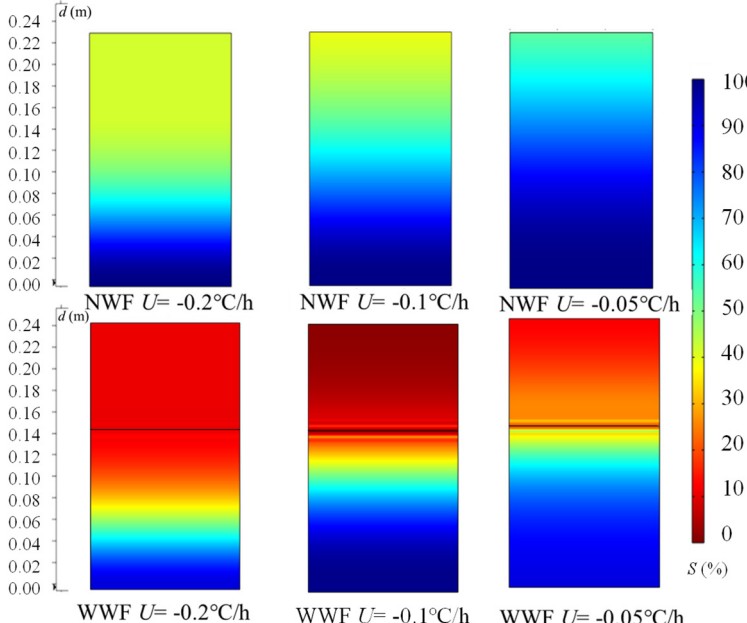

**Figure 15.** Comparison of saturation distribution between NWF model and WWF model in the freezing process.

On the other hand, though the WF is evidently capable of effectively draining water under different cooling rates, as shown in Figure 15, it can sufficiently remove moisture from the soil, especially at the low cooling rate. Consequently, low cooling rates result in a more apparent reduction in frost heave strains of the WWF model during the freezing process. For example, the saturation distribution in the soil is almost similar when the cooling rates are 0.1 °C/h and 0.2 °C/h. The saturation distributions of $U = 0.1$ °C/h and $U = 0.2$ °C/h in their corresponding NWF models are significantly different, indicating that the WF has a greater drainage effect when the cooling rate is lower.

## 6. Conclusions

The findings from this study can be summarized as follows:

(1)   A THM coupled FE model, which can simulate the freezing process of unsaturated soil, was combined with an evaporation model to evaluate the WF inhibition effect of frost heave under different situations. The proposed model can describe the influence of the surrounding environment, such as the saturation and temperature of the soil, on WF.

(2)   To examine the validity of the proposed model, the simulation results were compared to the experimental results. As a result, it was verified that the proposed model could sufficiently predict the temperature, moisture, and frost heave of the soil column in the frost heave test, and that the proposed model could simulate the suppression effect of WF on frost heave under different conditions.

(3)   The results from both the experimental and numerical simulations demonstrate that WF can effectively inhibit the occurrence of frost heave. This indicates that the proposed model is able to reproduce the transient evaporation process of WF during freezing, whose rate changes in accordance with the soil saturation and temperature, and that it can simulate the inhibition effect of WF on frost heave under different conditions.

(4)   The soil type, GWL, and cooling rate affect the inhibition effect of WF on frost heave strain in frost-susceptible soils. The WF was more effective in reducing frost heave on sandy soils than on clay soils. For the same freezing time, a higher freezing rate tends to cause less frost heave, and the inhibition effect of WF on frost heave is correspondingly weakened. On the other hand, when the GWL is increased, it also reduces the effectiveness of WF in preventing frost heave.

**Author Contributions:** Conceptualization, supervision, funding acquisition, and writing—review and editing, T.I.; methodology, software, validation, and writing—original draft, Y.W. funding acquisition, K.M. and C.U.; experiment conduction, T.Y. and S.O. All authors have read and agreed to the published version of the manuscript.

**Funding:** This research was funded by Grant-in-Aids for Scientific Research, grant numbers (A) (16H02360) and (B) (17H03307) from the Japan Society for the Promotion of Science (JSPS) KAKENHI. The authors also acknowledge the financial support from the China Scholarship Council (201907565040).

**Institutional Review Board Statement:** Not applicable.

**Informed Consent Statement:** Not applicable.

**Data Availability Statement:** Data sharing is not applicable.

**Acknowledgments:** This research was supported, in part, by Grant-in-Aids for Scientific Research (A) (16H02360) and (B) (17H03307) from the Japan Society for the Promotion of Science (JSPS) KAKENHI. The authors also acknowledge support from the China Scholarship Council.

**Conflicts of Interest:** The authors declare no conflict of interest.

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
