# Peer review of "Modeling Wicking Fabric Inhibition Effect on Frost Heave"

_applsci, doi:10.3390/app12094357_

Round 1

Reviewer 1 Report

Dear Authors,

My opinion is that the paper is very quality but must  be improved.

The suggestions are as follows:

  • What is the abbreviation „THM“? It is mentioned several times in the text but isn't explained.
  • The sentences of the above equation should end in a unique way. In some parts the sentences end with , (comma) and in some parts with : (colon).

:  -Line 115, 185, 200

,   - Line 125, 134, 138, 162, 165, 178, 190, 193, 195, 217, 232, 239, 242, 246, 247

  • Line 119 -  What it means „saturated-unsaturated“?
  • Line 237- Full stop is missing at the end.
  • Table 1- I think zhe letter „m“ is missing in „Van-Genuchten- Muale fitting parameter”.
  • Figure 6- Some curves are not visible at all. The curves should be corrected.
  • Line 399 – I think that is full stop better that semicolon.
  • Line 471- Full stop is missing after “as positive”.
  • It is recommended that when enumerating labels in equations, labels be explained one below the other.
  • Figure 12- Letters and parentheses overlap.
  • Line 488-491 – The sentence is confusing. It would be better to divide one sentences into several sentences.
  • The text in conclusion should be aligned (Justify option).

:  -Line 115, 185, 200

,   - Line 125, 134, 138, 162, 165, 178, 190, 193, 195, 217, 232, 239, 242, 246, 247

None    –Line 181

  • Line 119 -  What it means „saturated-unsaturated“?
  • Line 237- Full stop is missing at the end.
  • Table 1- I think zhe letter „m“ is missing in „Van-Genuchten- Muale fitting parameter”.
  • Figure 6- Some curves are not visible at all. The curves should be corrected.
  • Line 399 – I think that is full stop better that semicolon.
  • Line 471- Full stop is missing after “as positive”.
  • It is recommended that when enumerating labels in equations, labels be explained one below the other.
  • Figure 12- Letters and parentheses overlap.
  • Line 488-491 – The sentence is confusing. It would be better to divide one sentences into several sentences.
  • The text in conclusion should be aligned (Justify option).

Author Response

We sincerely thank the editor and the reviewers for your examination of this manuscript. The valuable comments from the editor and the reviewers are very helpful for us to revise and improve this manuscript. Based on the editor’s and reviewers’ comments, we revised the manuscript, and the revised parts are marked in the revised manuscript. Please kindly go through our responses on the attachment.

Reviewer 2 Report

The main objective of this manuscript is to experimentally study the effect of WF inhibition on frost under different conditions. This manuscript needs some modifications before it can be accepted for publication as follows:

  • The English language must be improved throughout the manuscript. For examples:
    • Line 19, “the depression effects”. The common expression in engineering must be used, such as “inhibition effects”.
    • Lines 20 and 103, “THM”, do you mean “Coupled Hydrothermal Mechanical (THM)”? The definition of each abbreviation must be mentioned the first time it is mentioned, either in the abstract or in the rest of the manuscript.
    • Lines 71 "Zhang and Presler (2012)", 73, 74, 77, 79, 81, 85, 88, 187, 192, 212, 230, 239, etc. The bracket containing the published year (2012) must be deleted. Citing references within the text and list of references must be compatible (the same style: Numeric or alphabetic) and follow the journal’s requirements.
    • The above comments apply to the whole manuscript.
  • Line 345, “850 quadrilateral cells”! Mesh sensitivity must be added.
  • The section “Properties and Standards of Materials” belongs to the experimental work (section 3) and not section 4. It must be moved to section 3
  • Line 404, [42]. Is the comparison between the present results or the previous results?
  • The conclusion should be reduced.

Author Response

(The authors gave the same response as above.)

Reviewer 3 Report

The manuscript is well presented with adequate data worthy of publication. The introduction is well written with a good number of reference papers cited and the conclusions are clear. I recommend this paper for publication after some minor revisions.

Author Response

We thank you for the time you put into reviewing our paper and look forward to meeting your expectations.

Reviewer 4 Report

Interesting study on the use and modelling of the WF used in pavement structure to limit frost heave. The paper is well written. The literature review is complete enough and shows the gaps in the actual knowledge. However, it would be useful to have a bit more information on how the water is drained from the WF in the pavement so it does not remain saturated (part of that is in section 2.3).

The objective is clear, but the methodology is explained too briefly. Several equations are presented in the paper and the explanation is short, but efficient.

I would like more info on why the water removed from WF is considered an evaporation process. In 2.3, it is mentioned the interaction of WF with air. In the context used here, If the WF is extended outside the pavement to have a contact with air (not well shown on Fig 1), it should be precisely stated.

Here are a few specific comments

  • Need to define THM; it’s not defined anywhere in the text
  • L29-31. Seems to be punctuation missing or the sentence is incomplete
  • L131-132: there’s a link missing between those 2 paragraphs.
  • L289: why 90% saturation?
  • L301: I think the columns size is good, but any reason why those heights and diameters were selected?
  • Fig 3: How did you make sure that water could not simply pass between the wall of the column and the WF?
  • L329: dropped rapidly – how fast?
  • L333: why this specific freezing range?
  • Figure 6: single test or average of several repetitions? Same questions for all the results presented.
  • For the different saturation figures, I would use the same order (NWF in the upper part) and I would indicate clearly the position of the WF

Author Response

We sincerely thank the editor and the reviewers for their examination of this manuscript. The valuable comments from the editor and the reviewers are very helpful for us to revise and improve this manuscript. Based on the editor’s and reviewers’ comments, we revised the manuscript, and the revised parts are marked in the revised manuscript. Please kindly go through our responses on the attachment.

Round 2

Reviewer 2 Report

The authors have successfully addressed all my comments. Therefore, I recommend the publication of this manuscript.